# Broken Tokens? Your Language Model can Secretly Handle Non-Canonical Tokenizations

Brian Siyuan Zheng [1]  Alisa Liu [1]  Orevaoghene Ahia [1]  Jonathan Hayase [1]  Yejin Choi [2]  Noah A. Smith [1 3]

## Abstract

Modern tokenizers employ deterministic algorithms to map text into a single "canonical" token sequence, yet the same string can be encoded as many non-canonical tokenizations using the tokenizer vocabulary. In this work, we investigate the robustness of LMs to text encoded with non-canonical tokenizations entirely unseen during training. Surprisingly, when evaluated across 20 benchmarks, we find that instruction-tuned models retain up to 93.4% of their original performance when given a randomly sampled tokenization, and 90.8% with character-level tokenization. We see that overall stronger models tend to be more robust, and robustness diminishes as the tokenization departs farther from the canonical form. Motivated by these results, we then identify settings where non-canonical tokenization schemes can *improve* performance, finding that character-level segmentation improves string manipulation and code understanding tasks by up to +14%, and right-aligned digit grouping enhances large-number arithmetic by +33%. Finally, we investigate the source of this robustness, finding that it arises in the instruction-tuning phase. We show that while both base and post-trained models grasp the semantics of non-canonical tokenizations (perceiving them as containing misspellings), base models try to mimic the imagined mistakes and degenerate into nonsensical output, while post-trained models are committed to fluent responses. Overall, our findings suggest that models are less tied to their tokenizer than previously believed, and demonstrate the promise of intervening on tokenization at inference time to boost performance.[1]

[1]University of Washington [2]Stanford University [3]Allen Institute for AI. Correspondence to: Brian Zheng <zhengbr@cs.washington.edu>.

*Non-archival presentation at ICML 2025 Tokenization Workshop (TokShop)*, Vancouver, Canada. 2025.

[1]Code is available at https://github.com/

## 1. Introduction

Tokenizers segment text into a sequence of discrete tokens in the language model's (LM) vocabulary. Most of today's LMs use deterministic subword tokenization, which produces a single canonical token sequence for a given piece of text, and further, for each whitespace-delimited word. One commonly discussed limitation of this approach is that, by mapping byte strings to symbolic token IDs, the orthographic makeup of tokens is obscured to the LM (Provilkov et al., 2020; Edman et al., 2024). This can be especially harmful for LM understanding of numbers (Nogueira et al., 2021; Thawani et al., 2021; Singh & Strouse, 2024) and morphologically rich languages (Arnett & Bergen, 2025; Hofmann et al., 2021), and has motivated efforts to model text directly at the byte level (Clark et al., 2022; Xue et al., 2022; Wang et al., 2024b; Tay et al., 2022; Yu et al., 2023; Nawrot et al., 2023; Pagnoni et al., 2024; Ahia et al., 2024; Limisiewicz et al., 2024).

To shed more light on this perceived limitation, in this work we study whether LMs can adapt *at inference time*, without any additional training, to a different tokenization scheme than the one they were trained with. While the tokenizer deterministically outputs a *canonical tokenization* of any text into tokens (usually by applying an ordered list of merge rules), *non-canonical tokenizations* of the same text using the same vocabulary are generally possible (see example in Figure 1). Here, we evaluate how LMs trained with deterministic tokenizers behave when given non-canonical tokenizations of text. Surprisingly, we find that *instruction-tuned LMs across many model families are extremely robust* to non-canonical tokenizations (section 2). For example, when evaluated across 20 benchmarks, `Qwen-2.5-7B-Instruct` retains 93.4% of its original performance when presented with a random non-canonical tokenization, and 90.8% when presented with character-level tokens (see Figure 1). Thus, far from not understanding the makeup of their tokens, LMs are able to compose token sequences in entirely new ways at inference time (Kaplan et al., 2025).

This leads to an intriguing possibility: if LMs can process

`Brianzhengca/Tokenizer-Robustness.`

*Figure 1.* **Left:** An example of how `Llama-3.1-8B-Instruct` responds when given **canonically tokenized** input, versus a **random tokenization** and **character-level tokenization**. The responses are surprisingly similar, demonstrating their ability to handle non-canonical tokenizations. In particular, LMs generally respond with correctly tokenized output regardless of the tokenization scheme used for the context. **Right:** Performance of `Qwen-2.5-7B-Instruct` across various benchmarks and segmentation strategies. The model preserves much of its original performance when presented with non-canonical tokenizations. Note that even when presented with non-canonical tokenizations, models produce outputs that are canonically tokenized.

non-canonical tokenizations, can we use different tokenization schemes at inference time to improve performance? For instance, prior work has found that better segmentation of large numbers can improve accuracy on arithmetic (Singh & Strouse, 2024; Sathe et al., 2025). Indeed, we identify several settings where non-canonical tokenization schemes improve performance for `Llama-3.1-Instruct` (section 3): character-level tokenization brings up to +14% improvement on string manipulation and code understanding tasks, perhaps by granting LMs more direct access to orthographic cues. Meanwhile, right-aligned digit groups, which provide a consistent grouping of digits by powers of a thousand, improves arithmetic on large numbers by +33%. These performance gains are achieved without any model finetuning, pointing to the promise of tokenization as a means of inference-time control.

Finally, we investigate the origins of model robustness to non-canonical tokenizations (section 4). Across multiple model families, we find that *pretrained-only* LMs consistently fail to produce fluent continuations given non-canonically tokenized context. By studying models at different stages of post-training, we identify that robustness arises during the supervised instruction-tuning (SFT) phase (subsection 4.1). We then ablate differences between pre-training and SFT procedures and find that the separation of the instruction and response as distinct turns of conversation is key (subsection 4.2). From here, we provide evidence for

a plausible explanation: while both base and post-trained models grasp the semantics of non-canonical tokenizations, they also perceive them as containing misspellings (subsection 4.2). Base models attempt to mimic the imagined mistakes and degenerate into nonsense, whereas post-trained models are not bound by the style of the instruction and thus able to produce fluent responses.

Overall, despite being trained with deterministic tokenization, instruction-tuned LMs readily accommodate new tokenizations at inference time, suggesting that LMs are less constrained by their tokenizer than previously believed (Minixhofer et al., 2024). Moreover, in settings where different representations of text are beneficial, we can intervene on tokenization at inference time for performance gains. We hope our work sheds new light on the discussion of strengths and limitations of tokenization, and points to the possibility of dynamically finding the optimal representation of text after pretraining.

## 2. Language Models are Robust to Non-Canonical Tokenizations

In our main experiments, we evaluate the robustness of LMs to non-canonical tokenizations by comparing their performance on downstream tasks when given different tokenizations of the input.

*Table 1.* **Evaluated across many benchmarks, models are surprisingly robust to non-canonical tokenizations of the context.** We show the absolute drop in performance when given a randomly sampled non-canonical tokenization (**Rand** $\Delta$) and character-level tokenization (**Char** $\Delta$), relative to the canonical (**Canon**) tokenization. We also summarize the model's ability to retain performance across benchmarks and tokenization strategies (bottom).

| Benchmark | QWEN-2.5-7B-INSTRUCT | | | LLAMA-3.1-8B-INSTRUCT | | | OLMO-2-7B-INSTRUCT | | |
|---|---|---|---|---|---|---|---|---|---|
| | Canon | Rand $\Delta$ | Char $\Delta$ | Canon | Rand $\Delta$ | Char $\Delta$ | Canon | Rand $\Delta$ | Char $\Delta$ |
| *Multiple choice (MC)* | | | | | | | | | |
| ARC-C | 86.4 | −1.80 | −2.60 | 76.2 | −14.10 | −22.40 | 77.0 | −37.40 | −44.60 |
| ARC-E | 94.4 | −2.20 | −3.60 | 91.3 | −12.60 | −21.50 | 85.4 | −37.00 | −49.00 |
| COPA | 97.0 | −1.80 | −7.40 | 97.2 | −9.60 | −14.80 | 93.8 | −21.40 | −33.60 |
| Winogrande | 46.0 | −4.60 | −3.00 | 59.6 | +2.00 | −5.00 | 58.6 | −7.80 | −7.00 |
| Winograd | 72.4 | −16.80 | −26.60 | 74.4 | −9.40 | −13.00 | 72.4 | −8.60 | −19.00 |
| CSQA | 85.6 | −8.60 | −9.20 | 77.6 | −11.40 | −20.00 | 75.4 | −31.60 | −40.00 |
| OpenbookQA | 87.2 | −7.00 | −7.80 | 82.0 | −13.80 | −20.20 | 76.2 | −30.80 | −40.80 |
| PIQA | 87.0 | −6.00 | −14.00 | 84.0 | −12.60 | −18.40 | 78.2 | −17.40 | −25.00 |
| MMLU | 71.7 | −3.70 | −7.30 | 68.2 | −11.60 | −24.00 | 59.5 | −16.30 | −29.10 |
| BoolQ | 84.8 | −5.00 | −4.80 | 86.2 | −19.20 | −17.20 | 71.0 | −4.00 | −9.20 |
| HellaSwag | 79.6 | −6.20 | −7.40 | 68.6 | −14.20 | −23.40 | 68.0 | −26.80 | −39.80 |
| *Short answer (SA)* | | | | | | | | | |
| WikidataQA | 81.2 | −7.60 | −9.00 | 78.6 | −12.40 | −18.00 | 73.2 | −28.80 | −32.20 |
| TOFU | 77.8 | +0.00 | −1.70 | 82.1 | +1.70 | +0.80 | 82.9 | −12.80 | −23.90 |
| TriviaQA | 70.0 | −1.00 | −3.80 | 76.6 | −9.80 | −13.60 | 70.0 | −22.20 | −34.80 |
| JeopardyQA | 49.4 | −5.60 | −8.00 | 43.6 | −2.20 | −10.20 | 42.6 | −21.60 | −24.20 |
| AlpacaEval | 50.0 | −3.70 | −1.80 | 50.0 | −5.70 | −7.50 | 50.0 | −2.10 | −11.30 |
| MATH | 53.9 | −2.40 | −2.70 | 32.0 | −4.20 | −9.70 | 22.7 | −5.20 | −9.20 |
| CUTE | 70.0 | −4.90 | −8.80 | 68.0 | −11.10 | −15.30 | 55.3 | −10.20 | −5.70 |
| GSM8K | 87.3 | −6.10 | −8.50 | 82.0 | −11.70 | −16.00 | 73.9 | −23.10 | −35.80 |
| DROP | 88.2 | −0.80 | +0.00 | 88.8 | −0.60 | −5.00 | 77.0 | −5.60 | −7.60 |
| Avg MC Retention (%) | $92.4_{\pm5.97}$ | $89.2_{\pm9.33}$ | $85.6_{\pm6.86}$ | $76.8_{\pm7.99}$ | $71.2_{\pm14.7}$ | $59.2_{\pm17.8}$ | | | |
| Avg SA Retention (%) | $94.6_{\pm3.97}$ | $92.7_{\pm5.35}$ | $90.3_{\pm6.78}$ | $82.7_{\pm9.65}$ | $75.4_{\pm15.1}$ | $65.4_{\pm17.4}$ | | | |
| Avg Overall Retention (%) | $93.4_{\pm5.15}$ | $90.8_{\pm7.81}$ | $87.7_{\pm7.05}$ | $79.4_{\pm9.05}$ | $73.1_{\pm14.7}$ | $62.0_{\pm17.5}$ | | | |

## 2.1. Background

Most LMs today, and all the models we study, use the *Byte-Pair Encoding* (BPE) (Sennrich et al., 2016) algorithm for tokenization. The BPE tokenizer is learned by splitting a corpus of text into bytes, which form the initial vocabulary, then iteratively merging the most frequent pair of tokens into a new token that is added to the vocabulary. To encode a new text, it is split into bytes, and the learned merges are applied in the same order. As a result, a BPE tokenizer always produces the same token sequence for the same text. Further, because BPE tokens do not cross whitespace boundaries, the same whitespace-delimited word is always represented with the same token or token sequence.

A natural observation is that given a tokenizer vocabulary, there exist many token sequences that decode to the same text. For instance, *cat* could be tokenized as [`cat`], [`, cat`], [`, c, at`], [`, c, a, t`], etc. In general, the number of non-canonical tokenizations grows exponentially with the length of the text. Many previous works have argued that the probability of a string should be calculated as the sum of probabilities of all possible tokenizations (Cao & Rimell, 2021; Chirkova et al., 2023; Geh et al., 2024). However, less attention has been paid to how non-canonical tokenizations affect LMs in generative settings.

## 2.2. Setup

We consider two non-canonical tokenization schemes. (1) *Random tokenization* produces a tokenization (uniformly at random) from the set of tokenizations more granular than the canonical one. This can be achieved by recursively splitting individual tokens into a valid pair of tokens, similarly to (Sims et al., 2025); the pseudocode and a proof of correctness is provided in Appendix A. (2) *Character-level tokenization* decomposes the string into character tokens, i.e., using no subword token from the vocabulary. For text containing only English letters and punctuation (where each character is exactly one byte), this produces the most granular possible tokenization. We consider three models, `Llama-3.1-8B-Instruct` (Meta, 2024), `OLMO2-7B-Instruct` (OLMo et al., 2024), and `Qwen-2.5-7B-Instruct` (Qwen, 2025), which we

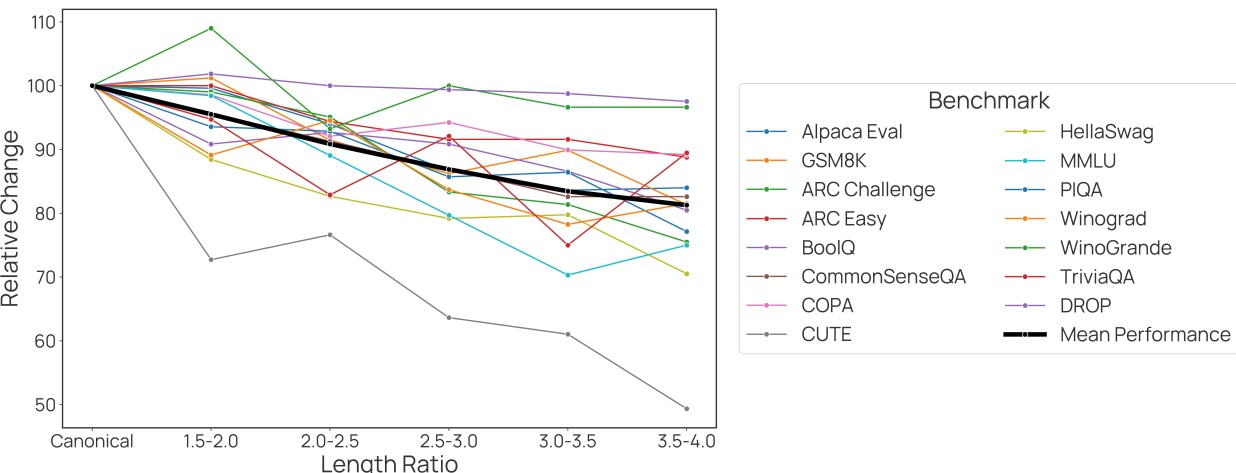

*Figure 2.* **Model performance generally declines as the tokenization becomes more granular.** We achieve variation in tokenization length using different values of $p$ in BPE-dropout, and group tokenizations into buckets based on how many times longer it is than the canonical tokenization.

evaluate on 20 benchmarks shown in Table 1. Please see subsection B.1 for further description of the datasets and evaluation setup.

### 2.3. Results

Shown in Table 1, while random tokenization consistently leads to worse performance compared to the canonical tokenization, the effect is small. On average across benchmarks, `Qwen-2.5` retains 93.4% of its performance when given random tokenization, followed by `Llama-3.1` at 87.7% and `Olmo-2` at 73.1%. The performance drops further with character-level tokenization, with retention of 90.8%, 79.4%, and 62.0% for the three models, respectively. This ranking of models in terms of retention is consistent with their ranking in absolute accuracy (under canonical tokenization), suggesting that stronger models are generally more robust to non-canonical tokenization strategies.

We also observe that all models retain performance better on short answer (SA) benchmarks (where the model generates an output in free-form) compared to multiple choice (MC) benchmarks (where the model is instructed to directly output the correct answer choice). In addition, LMs consistently *produce* correct token sequences even when conditioning on non-canonical tokenizations. We hypothesize that, in the SA setting, models benefit from eventually conditioning on recent correctly-tokenized context.

### 2.4. Analysis: How does granularity of the tokenization affect robustness?

We next study whether tokenization fine-grainedness correlates in general with model robustness. We measure the fine-grainedness of a given non-canonical tokenization by how many times longer it is (in tokens) than the canonical tokenization, which we call the "length ratio." Finer-grained tokenizations have higher ratios, while coarser ones have ratios closer to 1. We produce tokenizations with diverse length ratios by applying BPE dropout (Provilkov et al., 2020) with $p \in [0.1, 0.2, ..., 0.9]$, which controls the probability with which each merge is dropped. (High $p$ leads to finer-grained segmentations, and $p = 0.0$ corresponds to conventional BPE.)

Figure 2 shows the relationship between the length ratio and the average performance retention relative to canonical tokenization, with finer-grained tokenization generally leading to worse performance. When performance retention is averaged across tasks, the negative correlation is statistically significant under Kendall's $\tau$ with $p = 0.003$.

## 3. Can non-canonical tokenizations *improve* model performance?

If LMs can process non-canonical tokenizations, this points to the exciting possibility that tokenization schemes can be modified completely at inference-time. This would be useful if, in certain settings, there exists a better representation of text than what the tokenizer produces. In this section, we develop a suite of tasks that intuitively require understanding of the orthography of the text, and show that

*Table 2.* Examples from tasks we construct where non-canonical tokenizations lead to improved performance for `Llama-3.1-7B-Instruct` (section 3).

| |
|---|
| **Counting characters:** `Count the number of the letter 'r' in the word strawberry.` |
| **Acronyms:** `Come up with a sequence of words where the first letters would form this acronym: isman` |
| **Codeline Description:** `What does the following code do:` `{code block here}` `A. Counts paths from a point to reach Origin` `B. Program to check if a matrix is symmetric` `C. Longest subsequence from an array of pairs having first element increasing and second element decreasing .` `D. Count the number of strings in an array whose distinct characters are less than equal to M` |
| **Arithmetic:** `8492079913 + 4877278482 =` |

`Llama-3.1-8B-Instruct` performs better under non-canonical tokenization schemes.

### 3.1. Tasks

Please see Table 5 for an example question in each task and Table B.2 for further details on dataset construction. For all tasks except Arithmetic, we use character-level tokenization.

**Counting Characters** This task asks the model to count the number of occurrences of the most common letter in 5-10 character tokens in `Llama-3.1`'s vocabulary, and contains 1001 samples.

**Acronyms** This task asks models to generate a list of words whose first letters form a given acronym. We construct 3594 5-letter acronyms by sampling each letter uniformly at random from the alphabet.

**Code Description** For a more real-world application, we construct a task where the model is given a code snippet and asked to identify the function of the code in natural language from four MC options. The setup is inspired by the Codeline Description task from BIG-Bench (bench authors, 2023), but to increase the difficulty we use more complex code snippets and corresponding natural language descriptions from XLCoST (Zhu et al., 2022). To collect incorrect answers, we sample three other code descriptions from the dataset. This task contains 4800 samples across 6 programming languages.

**Arithmetic** Prior work has suggested that arithmetic is difficult for LMs in part due to poor segmentation of digits (Nogueira et al., 2021; Thawani et al., 2021). We curate a simple arithmetic dataset by constructing addition and subtraction tasks for 10 digit numbers. Here, we use a different segmentation strategy. The `Llama-3.1` tokenizer

segments numbers into groups of three left-to-right (e.g., *1000000* is encoded as ["100", "000", "0"]), due to the pre-tokenization regular expression looking for matches greedily from the left. Inspired by (Singh & Strouse, 2024), we instead segment digits into groups of three right-to-left (e.g., ["1", "000", "000"]). This task contains 1000 addition and subtraction questions in total.

### 3.2. Results

Shown in Table 3, in all the tasks we construct, the non-canonical tokenization strategy leads to substantially better performance compared to the canonical tokenization. In particular, we observe a +14.3% improvement on code description and +33.7% on arithmetic. Our results show that the tokenization scheme used in training is not necessarily the optimal one at inference-time, and replacing them with intuitively meaningful tokenizations can bring substantial performance gains. We leave automatically identifying the optimal tokenization as a promising direction for future work.

## 4. Investigating the Source of Robustness

Thus far, our experiments have used post-trained "instruct" models. In this section, we find that pretrained-only models are actually unable to produce fluent continuations of unusually tokenized context (subsection 4.1), perform ablations to identify the conditions enabling robustness (subsection 4.2), and finally provide support for an explanation of why generative robustness arises during post-training (subsection 4.3).

### 4.1. When does robustness appear in model training?

We first quantify the robustness of models at different stages of the model development pipeline by using the `Olmo2` and `Tulu3` (Lambert et al., 2025) model families which

*Table 3.* **On several tasks, `Llama-3.1-8B-Instruct` achieves substantially *better* performance when using a non-canonical tokenization scheme**. For the first three tasks, the input is tokenized at the character level; for Arithmetic, we segment digits into groups of three digits from right to left.

| Task | Canonical | Alternative | $\Delta$ |
|------|-----------|-------------|----------|
| Counting Characters | 66.5 | **73.5** | +6.99 |
| Acronyms | 49.7 | **57.4** | +7.74 |
| Code Description | 68.6 | **82.9** | +14.3 |
| Arithmetic | 36.5 | **70.2** | +33.7 |

include the base, SFT, DPO, and final instruct models. For simplicity, we focus on AlpacaEval and use character-level tokenization. For base models, we construct the prompt by placing the instruction in a question-answer template (`Question:` {instruction} `\nAnswer:`). We define three simple measures of generation quality.

**Spelling**  We measure the proportion of (whitespace-delimited) words in the generation that can be found in a collection of the top 10,000 most common English words.[2]

**Grammaticality**  We use LanguageTool's grammar checker[3] to count the number of grammatical mistakes, which we normalize by the number of words in the generation and subtract from 1 to produce a grammaticality score where higher is better.

**Win rate**  To measure overall generation quality, we use `alpaca_eval_gpt4` as an LM judge in the AlpacaEval framework and report the win rate of the generation given alternative against canonical tokenizations of the context. Unlike the previous two metrics, this measures not only the quality of the generation but also its relevance to the context.

Shown in Figure 3, the base models of `Olmo2` and `Llama-3.1` are both unable to produce sensible output conditioned on character-level tokenizations of context, scoring at best 0.317 on spelling and 0.260 on grammaticality. Qualitatively, generations are extremely difficult to parse and often involve odd character substitutions and repetitions (e.g., `Yoou`, `haviin`). Despite this, they sometimes reflect an understanding of the prompt. Consider, for example,

In contrast, the post-trained models are more robust across all three metrics, with *much of the improvement coming from the SFT stage alone*.

---

[2] https://github.com/first20hours/google-10000-english

[3] https://github.com/languagetool-org/languagetool

```
Question:  I like to host guests at my
home from time to time [...]
Can you give me a recipe for Canjeero?
Answer:   I aam glade tio hear tio hear
          tio hear tio hear that yoou enjoy
          haviin gauests at yoour hoome an tio
          keeep tio keeep tio keeep
```

### 4.2. Why do instruction-tuned models become robust?

We first replicate the finding from subsection 4.1 that SFT yields robustness to non-canonical tokenizations by fine-tuning the `Llama-3.2-1B` base model on the `Tulu 3 SFT Personas Instruction Following` dataset. Then, we perform the following interventions on the SFT training data and procedure to shed light on the possible source.

**Gradient over full sequence**  SFT on instruction-response pairs conventionally uses a loss mask over the instruction tokens, so that only the response tokens contribute to the loss. We remove this loss mask and instead compute gradients over the entire instruction and response.

**Question/answer template**  We replace the chat template with a simple question-answer template, `Question:` {instruction} `Answer:` {response}, both for training and evaluation.

**Removing the chat template**  We remove the chat template by concatenating the instruction and response without any special formatting. In evaluation, we again provide the instruction alone.

**Removing the instruction**  After SFT training, the LM's goal is no longer to continue a given text prefix, but rather to generate a response to the given instruction. To ablate the nature of the data itself, we take only the responses from the SFT data, and randomly split each into a new "prompt"

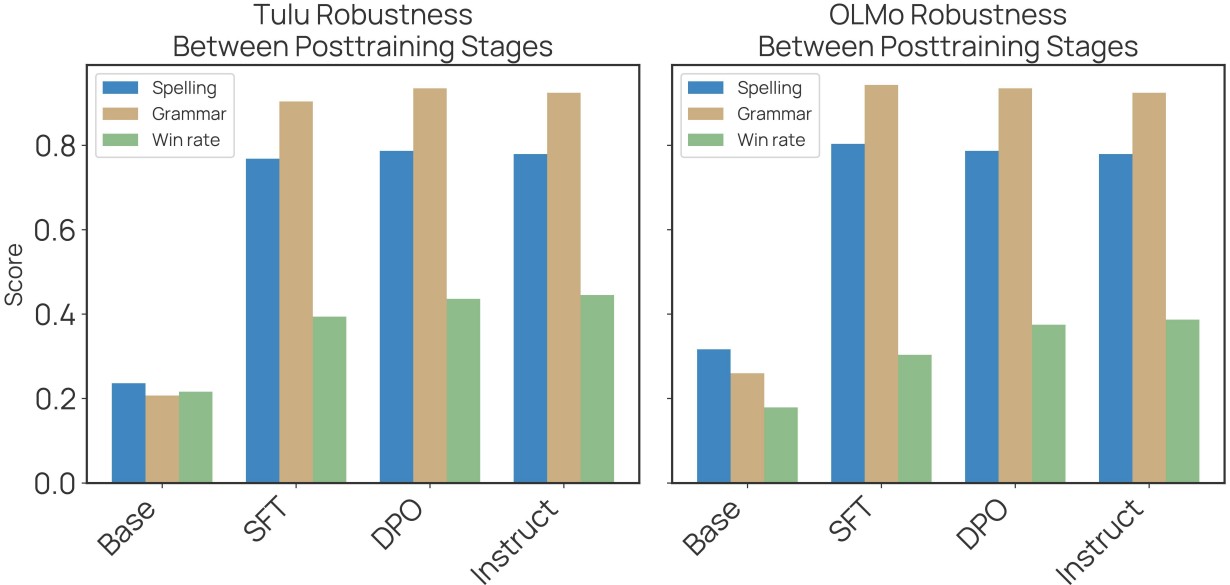

*Figure 3.* **Pretrained-only models completely fail to generate coherent output conditioned on non-canonical tokenizations of context; robustness is gained in the SFT stage.** We evaluate the spelling, grammaticality, and AlpacaEval win rate of model generations. Note that since `Tulu3` uses `Llama-3` as the base model, its base scores are computed using `Llama-3`'s base model scores.

and "response," which we format with the SFT template.[4] At test time, we similarly provide an incomplete response within "instruction" tags. Since the purpose of the passage is generally inferrable from the first few words of the gold response ("*Sure, here's a recipe for Kubdari...*"), we are able to evaluate generated responses under the same AlpacaEval framework.

Our results are shown in Figure 4. We replicate the finding that SFT (**No ablation**) leads the model to be able to handle non-canonical tokenizations. This persists when computing gradients over the entire instruction and response (**Full gradient**) so that the training procedure matches regular pretraining. Replacing the original chat template with a simple question-answer template (**QA template**) also maintains model robustness. However, the usage of a template is crucial — when directly concatenating the instruction and response (**Removing chat template**), the model fails to produce coherent generations, with the spelling score dropping from 0.786 in the no ablation setting to 0.0698. Inserting the chat template into pretraining-style data (**Removing the instruction**) also does not yield robustness, with a spelling and grammaticality scores remaining low at 0.181 and 0.158, respectively. Overall, these findings suggest that in order for the LM to generate fluent continuations given non-canonical tokenizations, the context and expected continuation need

---

[4]We match the instruction length distribution by counting the number of tokens $n$ in the original instruction, and formatting the first $n$ tokens of the response as the new "instruction."

to represent separate turns of dialogue, and additionally, be demarcated with a special template.

### 4.3. Disentangling understanding from generation

One plausible explanation for our findings thus far is that both base and instruction-tuned models grasp the semantics of non-canonical tokenizations, yet falsely perceive them as containing misspellings. While base LMs attempt to faithfully continue these mistakes and degenerate into nonsensical output, instruction-tuned models are trained to provide fluent responses regardless of the instruction, leading to the results observed in subsection 4.1. To test this hypothesis, we construct two simple tests:

1. **Word Repeat**: To determine if a model perceives the meaning of a word with non-canonical tokenization, we prompt the model to repeat a given word (while correcting any typos).

2. **Identifying Misspellings**: To determine if a model perceives a misspelling, we ask it to identify the word with a misspelling among two options: a (correctly tokenized) misspelling of a word and an non-canonical tokenization of that word (correctly spelled).

Results are shown in Table 4. Consistent with our hypothesis, we find that both the base and instruct models from the `Llama-3.1-8B` family score highly ($> 90\%$) on Word

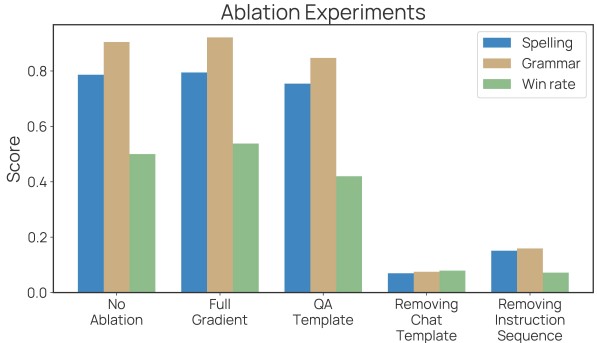

*Figure 4.* Ablations on the SFT training data and procedure indicate that the separation of the context and expected continuation — as different turns of dialogue demarcated with a special token — is key to robustness to non-canonical tokenizations.

**Repeat.** This means that the base model, despite its poor performance in subsection 4.1, actually recognizes the correct form of non-canonical tokenizations as well as its post-trained counterpart. In addition, both models perform at random when asked to distinguish non-canonical tokenization from true misspellings. In other words, the instruct model produces fluent responses (section 2) while interpreting the instruction as heavily misspelled! While instruct models evidently overcome this, the base model likely attempts to mimic the (perceived) idiosyncratic surface form, thus producing nonsensical (yet sometimes relevant) outputs.

*Table 4.* Both base and instruct models from the `Llama-3.1` family recognize words represented with non-canonical tokenizations (performing well on **Word Repeat**), but incorrectly perceive that there are misspellings (performing at random on **Identifying Misspellings**).

|  | Word Repeat | Identifying Misspellings |
|---|---|---|
| Llama-3.1-8B | 90.8 | 48.2 |
| Llama-3.1-8B-Instruct | 92.0 | 55.8 |

## 5. Related Work

**Robustness to tokenization** The extent to which LMs are limited by their tokenization is a topic of much debate, with the story evolving as LMs become larger and more capable. It is commonly argued that tokenization obscures orthographic information about tokens from the LM, leading to unexpected failures (Edman et al., 2024; Chai et al., 2024; Wang et al., 2024a). As a result, there have been many efforts towards linguistically-informed tokenization that make derivational, compound, and morphological boundaries within words explicit (Klein & Tsarfaty, 2020; Hofmann et al., 2021; 2022; Yehezkel & Pinter, 2023; Bauwens & Delobelle, 2024). Similarly, BPE-dropout (Provilkov

et al., 2020) and related methods (Sims et al., 2025) introduce variation in how a given string is tokenized to make models more robust to rare, misspelled, and unseen words.

However, other evidence suggests that LMs naturally overcome these limitations. For instance, token embeddings have been found to robustly encode character-level information, especially in larger models (Kaushal & Mahowald, 2022; Itzhak & Levy, 2022). This may be because word variants that do not share tokens in common (consider e.g., [dictionary] and [diction, aries], as tokenized by GPT-2) incentivize the model to learn spelling as a general solution to understanding their relations (Kaushal & Mahowald, 2022). Other works argue that LMs maintain an implicit vocabulary, and can compose arbitrary token sequences (including non-canonical ones) into useful higher-level representations (Feucht et al., 2024; Kaplan et al., 2025). Even in domains like biomedical text where terms are highly agglutinative, using tokenizers that segment on meaningful components does not lead to improved models (Jimenez Gutierrez et al., 2023). Recent works have even found that coarser *superword* tokenization (Liu et al., 2025; Schmidt et al., 2025), which capture common word sequences in a single token, preserve character-level understanding while providing benefits in compression and downstream performance.

Our work informs this conversation by showing that LMs can effectively leverage character-level knowledge of their tokens and glean potential benefits of improved representation at inference time.

**Non-canonical tokenizations** It has long been recognized that there are many possible ways to segment a string into tokens with a fixed vocabulary (Church, 2020), which in principle should be considered in the calculation of a string's likelihood (Cao & Rimell, 2021; Chirkova et al., 2023; Geh et al., 2024). In contemporaneous work, Geh et al. (2025) also show that LMs retain semantic understanding of non-canonical tokenizations. However, they only study this over a small set of 15 examples, as their main focus is to show that non-canonical tokenizations can be constructed adversarially to trigger unsafe completions. In contrast, we provide a more systematic study of LM robustness using benchmark evaluations and additionally study its source.

Somewhat relatedly, other works have provided algorithms for sampling at the character- or byte-level from tokenizer-based LMs (Phan et al., 2024; Vieira et al., 2024; Athiwaratkun et al., 2024; ?). Together, these directions suggest that despite being trained with one deterministic tokenization scheme, LMs can both condition on and produce token sequences over a different (sub)vocabulary.

# 6. Conclusion

Despite being trained with deterministic tokenization algorithms, we show that instruction-tuned language models are surprisingly robust to token sequences not seen in training. In certain domains, such as arithmetic or code, more intuitively meaningful tokenizations can even be swapped-in at inference time for improved performance. We analyze the source of this robustness, and find that while the base and instruct models both perceive the semantics of non-canonical tokenizations, only instruct models are capable of providing fluent continuations. Our work demonstrates a way in which LMs are not necessarily tied to tokenizer they were trained with, and highlights the potential of finding more optimal representations of text after pretraining.

ACKNOWLEDGMENTS

We would like to thank Joseph An and Ricky Koppolu, as well as the broader UW NLP community, for helpful conversations about this work. AL and JH are supported by the NSF Graduate Research Fellowship.

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

# A. Random Non-Canonical Tokenization

In this section, we provide our algorithm for producing a random non-canonical tokenization, and a proof that each non-canonical tokenization that is *more fine-grained* than the canonical one has equal probability of being output.

## A.1. Algorithm

We will split each token in a canonical tokenization into smaller tokens (that each exists in the tokenizer's vocabulary). We formulate our problem as: Given a valid token $t$, and a set of vocabulary $\mathcal{V}$, construct a sequence of tokens $seq$ using tokens that exist in $\mathcal{V}$ and form $t$ when concatenated together. We produce $seq$ using a recursive algorithm. Since there can be many possible $seq$ for each $t$, we need to randomly choose one and guarantee that each possible $seq$ is chosen with equal probability. We achieve this by considering recursion as producing a tree. Each path down the tree corresponds to one possible way to segment $t$. Each node of the tree represents a segmentation state where we have chosen some number of sub-tokens. At each node, we weigh the choice of which child node to visit by the number of leaves in the sub-tree that is rooted at each child node. This guarantees that each path down the tree is chosen with equal probability since the number of paths down a tree is equal to the number of leaf nodes in that tree. The pseudocode for the algorithm is in 1.

---

**Algorithm 1** Random Token Segmentation

---

0: **function** COUNTSEGMENTS($start$) {Cached (using memoization)}
0:     **if** $start = |token|$ **then**
0:         **return** 1 {Reached end; valid segmentation}
0:     **end if**
0:     $total \leftarrow 0$
0:     **for** $end \leftarrow start + 1$ **to** $|token|$ **do**
0:         $substring \leftarrow token[start : end]$
0:         **if** $substring \in vocabulary$ **then**
0:             $total \leftarrow total + $ COUNTSEGMENTS($end$)
0:         **end if**
0:     **end for**
0:     **return** $total$
0: **end function**
0: **function** BUILDSEGMENTS($start$)
0:     **if** $start = |token|$ **then**
0:         **return** $\varnothing$ {Empty segmentation}
0:     **end if**
0:     $validSegments \leftarrow []$
0:     $weights \leftarrow []$
0:     **for** $end \leftarrow start + 1$ **to** $|token|$ **do**
0:         $substring \leftarrow token[start : end]$
0:         **if** $substring \in vocabulary$ **then**
0:             $segCount \leftarrow $ COUNTSEGMENTS($end$)
0:             **if** $segCount > 0$ **then**
0:                 Append $substring$ to $validSegments$
0:                 Append $segCount$ to $weights$
0:             **end if**
0:         **end if**
0:     **end for**
0:     **if** $validSegments$ is empty **then**
0:         **return** $\varnothing$
0:     **end if**
0:     $chosenSegment \leftarrow $ weightedRandomChoice($validSegments, weights$)
0:     **return** $[chosenSegment] \parallel$ BUILDSEGMENTS($start + |chosenSegment|$) {Concatenate chosen segment with segmentation of remaining token}
0: **end function**
0: **procedure** SEGMENTTOKEN($token, vocabulary$)
0:     **if** COUNTSEGMENTS($0$) $= 0$ **then**
0:         **return** $\varnothing$ {No valid segmentation exists}
0:     **else**
0:         **return** BUILDSEGMENTS($0$)
0:     **end if**
0: **end procedure**=0

---

## A.2. Proof

**Goal:** To prove that the random segmentation algorithm chooses one valid segmentation from all possible valid segmentations with uniform probability.

**Notation:** Let $W(i)$ denote the number of valid segmentation completions (i.e., the number of leaves in the recursive tree) for the substring starting at index $i$. In particular, $W(|token|) = 1$. Note that $W(i)$ is calculated by the memoized recursive function $countSegments(i)$, which calculates the number of leaves of the subtree rooted at $i$.

**Base Case:** Consider the node corresponding to $i = |token|$ (the end of the token). Here, there is exactly one valid segmentation (the empty segmentation), so the algorithm returns it with probability 1. That is, every segmentation (in this case, the only one) is chosen with probability

$$\frac{1}{W(|token|)} = \frac{1}{1} = 1.$$

Thus, the base case holds.

**Inductive Hypothesis:** Assume that for any node corresponding to an index $j$ with $j > i$ (i.e., deeper in the recursion tree), every complete segmentation (leaf) in the subtree rooted at $j$ is chosen with probability

$$\frac{1}{W(j)}.$$

**Inductive Step:** Now consider a node corresponding to index $i$ (with $i < |token|$). Suppose that from $i$ there are $k$ valid branches corresponding to choosing substrings that end at indices $j_1, j_2, \ldots, j_k$, where for each $j$ we have $i < j \le |token|$ and the substring $token[i : j]$ is in the vocabulary. By definition,

$$W(i) = \sum_{r=1}^{k} W(j_r).$$

The algorithm selects the branch from $i$ to a specific child $j$ with probability

$$P(i \to j) = \frac{W(j)}{W(i)}.$$

Once branch $i \to j$ is chosen, by the inductive hypothesis every complete segmentation (leaf) in the subtree rooted at $j$ is chosen with probability

$$\frac{1}{W(j)}.$$

Thus, the probability $P(S)$ of obtaining a particular complete segmentation $\mathcal{V}$ that starts at $i$ by first taking the branch $i \to j$ and then following a specific path in the subtree rooted at $j$ is

$$P(S) = \frac{W(j)}{W(i)} \cdot \frac{1}{W(j)} = \frac{1}{W(i)}.$$

Since the factor $W(j)$ cancels, the probability $P(S)$ is independent of the particular child $j$ chosen.

**Conclusion:** By the inductive step, every complete segmentation (leaf) in the subtree rooted at any index $i$ is chosen with probability $\frac{1}{W(i)}$. In particular, when $i = 0$ (the start of the token), every valid segmentation of the entire token is selected with uniform probability $\frac{1}{W(0)}$. This completes the proof.

## B. Evaluation Details

### B.1. General benchmarks

For short-answer benchmarks, the system prompt is:

```
You are a helpful assistant.
```

For multiple-choice benchmarks, the system prompt is:

```
You are a helpful assistant. For the following multiple choice questions,
return the answer only, without any additional reasoning or explanation.
```

**MATH**   MATH is a dataset composed of fairly difficult, competition level math problems (Hendrycks et al., 2021b). The test set is composed of short answer problem that describe some scenario and asks the model to output a mathematically correct answer.

**GSM8K**    GSM8K is a dataset consisting of relatively simple math questions that would appear in grade school math exams (Cobbe et al., 2021). For GSM8K, the evaluations were done in the same manner as MATH.

**MMLU**    MMLU is a benchmarks comprising of multiple choice questions from a wide variety of subjects. (Hendrycks et al., 2021a) We sampled 500 questions from MMLU for our evaluation. We instructed the model to only output one answer to each question without any explanation.

**Alpaca Eval**    Alpaca Eval is an evaluation benchmark where generations from language models against given prompts are compared and judged by an annotator model. (Dubois et al., 2023) The metric used was raw winrate of the perturbed model as judged by a language model. The annotator we used was *alpaca_eval_gpt4*, which has been shown to have the highest Spearman and Pearson correlation coefficient with human annotators.

**ARC Challenge and ARC Easy**    Contains multiple choice questions with four options each, taken from grade school science exams (Clark et al., 2018). ARC Easy is tests basic science knowledge while ARC Challenge requires some procedural reasoning.

**BoolQ**    Contains true or false questions along with a context passage that provides the answer to the question. (Clark et al., 2019)

**CommonsenseQA**    Contains multiple choice questions with five options each that requires common sense knowledge to answer. (Talmor et al., 2019)

**COPA**    Contains multiple choice questions with two options each that tests knowledge of cause and effect. (Roemmele et al., 2011)

**CUTE**    Contains questions that require the model to manipulate sentence-level, word-level, and character-level structure for strings. (Edman et al., 2024)

**DROP**    contains questions that potentially require reasoning multiple pieces of information present in a given passage. (Dua et al., 2019)

**HellaSwag**    contains multiples choice questions with four options each that asks for the most natural continuation to some given context. (Zellers et al., 2019)

**JeopardyQA**    contains short answer questions from the "Jeopardy!" game show. (Kaggle, 2019)

**OpenbookQA**    contains multiple choice questions with four options each that require some multi-step and common sense reasoning. (Mihaylov et al., 2018)

**PIQA**    contains multiple choice questions that require reasoning about the physical world. (Bisk et al., 2020)

**TriviaQA**    contains short answer questions that requires knowledge of the world. (Joshi et al., 2017)

**Winograd**    contains multiple choice questions with two options that asks to determine what a pronoun might refer to. Answering these questions require knowledge of commen sense and surrounding context. (Levesque et al., 2012)

**Winogrande**    contains questions in the same format of Winograd but there are more questions and the questions are harder. (Sakaguchi et al., 2021)

**TOFU**    contains general short answer questions that tests the model's ability to process world knowledge. This is the retain set of the task of fictitious unlearning dataset. (Maini et al., 2024)

**WikidataQA**    require models to complete factual statements. (bench authors, 2023)

*Table 5.* System prompt for tasks in section 3. See Table 5 for example instructions.

| |
|---|
| **Counting characters:** You are a helpful assistant. The following prompt will ask you to return a sequence of words. Only return the sequence, separated by spaces. Do not provide any additional text or explanation. |
| 2) Choose the correct option (A, B, C, or D) from the provided choices. Your response must be a single letter: A, B, C, or D. Do not provide any additional text, explanation, or formatting unless explicitly requested. |
| **Code Description:** You are a programming assistant trained to analyze and interpret code snippets. When provided with a code snippet and a set of answer choices (A, B, C, or D), your task is to evaluate the code, determine its behavior, and select the answer that best describes this behavior. Your response must be a single letter: A, B, C, or D. Do not provide explanations or additional text unless explicitly requested. |
| **Arithmetic:** You are a computational assistant trained to evaluate arithmetic operations. When provided with an arithmetic expression, calculate the result and round it to the nearest integer. Respond only with the rounded result, without any additional text or explanation. |

### B.2. Constructed Benchmarks

In this section, we provide more detail on how datasets we use in section 3 are constructed.

**Count Characters Task**   The prompt asks the model to count the number of occurrences of a given character in a 10-character word; we always use the most frequently occurring character. Evaluation was done, similar to GSM and MATH, by finding the last number in the generated response. Generations without any numbers are considered incorrect.

**Generate Acronym Task**   The model is asked to generate a sequence of words whose first letters form a randomly sampled five character string. For evaluation, we take the first character of each whitespace-delimited word and check if it matches the desired acronym.

**Codeline Description Task**   The model is asked to comprehend a piece of code and choose the best description from four options.

**Arithmetic Task**   The model is asked to perform addition or subtraction with 10 digit numbers. We use regex to extract numbers from the generation, which are then compared to the ground truth answer.

### B.3. Metrics of generation quality

Here we provide additional details on the metrics defined in subsection 4.1.

**Spelling**   We use the top 10000 most frequently appearing English words in Google's trillion word corpus. We only consider words with more than one character. This is because sometimes base models will repeatedly generate the same letter, and since all English letters are in the word list, the generation would receive a high score.

**Grammaticality**   One drawback with this evaluation method is that oftentimes the model would repeat the same letter over and over again, or start counting numbers. In both of these cases, there are no detected grammar mistakes, however they are still obviously gibberish. Therefore, we only calculate grammaticality scores for generations that receive a score $\geq 0.5$ on spelling; otherwise, we give it a grammaticality score of 0.

*Table 6.* Data format of ablations in subsection 4.2.

| |
|---|
| **No ablation:**  `<\|user\|>Provide a detailed analysis of Candace Parker's defensive techniques in her recent games, excluding the words "aggressive" and "blocking", in the format of a sports commentary script.  <\|assistant\|>[Sports Commentary Script] [Opening Scene...` |
| **QA Template:**  `Question: Provide a detailed analysis of Candace Parker's defensive techniques in her recent games, excluding the words "aggressive" and "blocking", in the format of a sports commentary script.  Answer:  [Sports Commentary Script] [Opening Scene...` |
| **Removing the chat template:**  `Provide a detailed analysis of Candace Parker's defensive techniques in her recent games, excluding the words "aggressive" and "blocking", in the format of a sports commentary script.  [Sports Commentary Script] [Opening Scene...` |
| **Removing the instruction:** `<\|user\|>[Sports Commentary Script] [Opening Scene:  A packed basketball arena, with fans eagerly awaiting the analysis of Candace Parker's recent performances on the court.] Commentator 1:  Welcome back, basketball fans!  <\|assistant\|>Tonight, we're diving into the defensive prowess of Candace Parker...` |

**Win rate**    Similar to evaluation in section 2, we also used `alpaca_eval_gpt4` as the evaluator and report raw win rate. In 4.1, the win rate is calculated against generations conditioned on input with canonical tokenization. In 4.2, the win rate is against generations from the **No Ablation** setting when also given character-level tokenization. By construction, the win rate of the **No Ablation** setting itself is 50%.

### B.4. Ablation Settings

For ablations on the data format, see examples of formatted data in Table 6.  Our finetuning code was forked from `allenai/open-instruct`.

### B.5. Disentangling understanding from generation

For these tasks, we use 500 words randomly sampled from Google's 10000 English word list[5].

**Word Repeat**    An example prompt is shown below.

```
Repeat each word directly, while correcting any typos.

Question: guarantees
Answer: guarantees

Question: revelation {Character-level tokenization}
Answer:
```

**Identifying Misspellings**    We obtain the misspelled word by randomly adding, removing, or substituting a single character from the word. An example prompt is shown below.

```
Question: Which of the two words contains a misspelling? Respond directly
with the answer option.

Question:
```

---

[5]https://github.com/first20hours/google-10000-english/blob/master/google-10000-english.txt

```
A. guarantees
B. garantees

Answer: B

{9 more in context examples}

Question:

A. farmer {Character-Level tokenization}
B. farme {Canonical tokenization}
```

