# OpenReview forum: "Broken Tokens? Your Language Model can Secretly Handle Non-Canonical Tokenizations"
_ICML.cc/2025/Workshop/TokShop — TokShop_

### Official Review · Reviewer_73Ti · 2025-06-07
**A very nice work on non-canonical tokenizations and the robustness of LLMs to them, complete with many experiments and a discussion of the underlying cause**

**Rating:** 8
**Confidence:** 5

**Review:**

This paper investigates the effect of non-canonical tokenizations on LLM inference. In particular, they cover random tokenizations and the character-level tokenization and show that, language models are (perhaps surprisingly) robust to the former, and can even benefit (depending on the task, and the tasks they choose are good ones for this) the latter. The authors evaluate their claims using 3 LLMs and a huge number of tasks. Overall, I enjoyed this paper and think it is a nice addition to the literature (and is timely, as several concurrent works have investigated the reliance of LLMs on their tokenization algorithm; Geh et al., Singh & Strouse, etc.).

I would vote to accept this paper.

Strengths:
- This paper investigates an important and timely problem
  - Their goal is to see how non-canonical tokenization affects LLM inference, and they do a good job of this
- The experiments cover several domains, several LLMs, and the different tokenization algorithms, so it is fairly comprehensive
- The motivation and findings are clear and well presented
- They additionally investigate where the robustness to tokenization comes from (i.e., is it present in pretraining? does it come later? etc.)
  - This is a very nice question to have asked, and adds a lot to the paper
  - I was surprised by the findings and likely others will be too

Weaknesses:
- The main weakness I see is that the LLMs used are all generally the same size (and fairly small)
  - Does this trend hold in LLMs that are an order of magnitude bigger?
  - How about much smaller? ~1B and lower?
- Which model(s) does Figure 2 refer to?
  - And, in this one, you use BPE-Dropout. Why not your random tokenization algorithm?

Nits:
- Use Title Case for Section/Subsection/Figure/etc. references
- [Line 159-160], I think you intended to have a subword marker in front of cat, maybe you did `\textvisiblespace` but didn't have it in math mode (or sometimes that doesn't show up in math mode either, in which case I use `$\texttt{\textvisiblespace})$`)
- While I do agree that LLMs are surprisingly robust to non-canonical tokenizations the following phrasing is a bit weird:
  - In Table 1 and Line 197, you show that, across the board, non-canonical tokenization has lower performance than canonical tokenization
  - You write "the effect is small" on line 197, but we can see in Table 1 that, for example, there are \delta >27% in some of the cases
  - On the other hand, in Table 3 and Line 268 (right side), you show that character-level tokenization does better
  - In this one, the \delta is relatively small compared to Table 1 (the average is ~14%), but you say "substantially better"
- Another reference for character-level knowledge in LLM embeddings is https://arxiv.org/abs/2402.09808
- You restrict your algorithm for random tokenizations to those that are "more granular than the canonical one"
  - I took this to mean "for each token in the canonical tokenization, generate a (uniformly) random re-segmentation of that token" and not "generate a tokenization uniformly at random"
  - A similar algorithm to yours exists for sampling at uniformly random from a DAG (and so also trees) which can be used to produce random tokenizations from the set of _all_ tokenizations
    - See https://repository.rit.edu/cgi/viewcontent.cgi?article=1666&context=theses (Chapter 5) and https://aclanthology.org/2024.emnlp-main.600/ (Section 4) and https://math.stackexchange.com/a/4084737/211815
  - I don't think this would change your results at all, but it is useful to think about this because you get a broader set of tokenizations, and, if you treat the tokenization as a DAG, you can directly sample tokenizations of a specific length (for example, for your results in Figure 2) by intersecting it with an automaton that accepts only strings of length `k`

Other:
- It's too bad you can't force your own tokenization in proprietary LLMs
  - Having the character-level encoding for math problems seems like a really useful thing to be able to do, but it depends heavily on the tokenizer (e.g. Llama2 automatically does it character-by-character, Llama3 chunks it, etc.).
  - How robust are models to changes in tokenization that are caused by changes in the surface form of the prompt?
    - e.g., if I forced a character level representation of a number by adding spaces between each digit "123 + 456" -> "1 2 3 + 4 5 6"
  - What about other strategies?

---

### Official Review · Reviewer_TB4J · 2025-06-08
**The paper "Broken Tokens" investigates the relationship between LMs and their tokenizers. The generally assumed narrative is that there is a tight coupling  between LMs and their deterministic tokenizer. Author(s) demonstrate that this is untrue and there is remarkable robustness to non canonical tokenizations i.e. alternative(and valid) segmentation of input that was not encountered during training. Author(s) rightly identify a set of tasks where the alternative test time tokenization can improve the performance namely string manipulation / coding and arithmetic tasks. They then try to identify the source of tokenization paradigm “resilience” in the model and present a case that the reason for resilience is the post-training of the model.**

**Rating:** 8
**Confidence:** 5

**Review:**

The papers contribution can be divided into three buckets.

1. Demonstrating Robustness to Unseen Tokenizations schemes.
This I believe is the core contribution of the work. The author(s) demonstrate that instruction-tuned LMs are highly resilient to different kinds of tokenizations at test time. To demonstrate this, they take three open source models and two different tokenization schemes (namely random tokenization and character level tokenization). Then they evaluate and benchmark the performance with respect to the conventional tokenization used in the model (which is BPE in most cases) across a suite of 20 tasks. The different tokenizations surprisingly perform very well, retaining up to 93.4%, which seems to point to the fact that LMs are highly resilient to changes in tokenization.

2. Pioneering usage of alternative Inference Time Tokenization Schemes to improve model performance.
This in my opinion is the most interesting part of the work. The author(s) identify a suite of tasks where alternative tokenization schemes at test time can help with performance. The two tasks that caught my eye were the code description and arithmetic tasks, which saw substantial improvements of +15.0% and +33.7% respectively. This can be good method for model control that is zero cost!
They are very important real world tasks and improvement shown is quite substantial.

3. Identifying the source robustness.
The author(s) then try to identify the source of this robustness. The work here seems somewhat preliminary, but the central hypothesis is insightful. The author(s) provide plausible reasons using clever ablation studies directionally pointing to the fact that the capability emerges during SFT stage. They argue that while base models perceive non canonical tokenization as misspellings, instruction tuned models prioritize the user's intent and produce a fluent response. Good finding.


In my opinion the strengths of paper.
1) The paper demonstrates LMs are quite resilient to changes in tokenization schemes during test time. The evaluation across three model families and 20 benchmarks seems quite exhaustive and supports the authors' hypothesis.
2) The paper correctly identifies the kinds of tasks where unconventional tokenization can help the model and can act as a lightweight paradigm for model control.
3) The paper provides a compelling even though preliminary explanation for the source of this resilience. This directionally points to the instruction tuning phase and offers a new fresh perspective on its function.


There are a few scope of improvements.
1) Ideally I would like author(s) to identify same tasks where models performance well in spite changes in test time tokenization for determining source of resilience. The analysis of the source of resilience could be strengthened by using the same downstream tasks for evaluation as were used to demonstrate the initial robustness. That would be more apple to apple comparison.
2) More real world usage tasks would be helpful. While some tasks are great like code comprehension, a few other tasks seem made up. I think some morphologically rich languages (For example, Turkish), where standard subword tokenizers struggle would be great tasks to measure performance.
3) The experiments are conducted a smaller ~8B parameter models. Does this finding hold for larger models? I am not sure and author(s) will have a more substantial contribution if they can try to replicate the findings for larger models.

---

### Official Review · Reviewer_M7wN · 2025-06-09
**Provides insights into unexpected robustness of LLMs with non-canonical token representation**

**Rating:** 7
**Confidence:** 3

**Review:**

## Key Contributions
The paper, "Broken Tokens? Your Language Model can Secretly Handle Non-Canonical Tokenizations," makes several significant contributions to the understanding of Large Language Models (LMs). It reveals the remarkable robustness of instruction-tuned models to non-canonical tokenizations, demonstrating that these models can retain a high percentage of their performance even when inputs are encoded in ways unseen during training. Furthermore, the research surprisingly shows that using certain non-canonical tokenization schemes at inference time can actually improve performance on specific tasks, such as enhancing string manipulation and code understanding with character-level segmentation, and boosting large-number arithmetic with right-aligned digit grouping, all without requiring additional fine-tuning. Finally, the study attributes this unexpected robustness primarily to the instruction-tuning phase. This is because instruction-tuned models, unlike their base counterparts, commit to generating fluent responses even with "misspellings" from non-canonical inputs, largely due to the structured separation of instruction and response during Supervised Fine-Tuning.

## Strengths
Soundness: The methodology appears sound, involving evaluation across 20 benchmarks to assess the performance of LMs under various non-canonical tokenizations. The algorithm and pseudocode presented demonstrate sound methodology, ensuring a uniform random distribution of tokens. The systematic approach to comparing base models with instruction-tuned models helps to isolate the source of the observed robustness. The quantitative results, such as the 93.4% performance retention and specific task improvements, provide concrete evidence for the claims.

Significance: The findings are highly significant for the field of natural language processing. By demonstrating that LMs are less dependent on their specific tokenizer than previously assumed, the paper opens up new avenues for dynamic tokenization strategies at inference time. This could lead to more efficient and specialized LM applications, particularly in areas like string manipulation, code understanding, and numerical tasks. The identification of instruction-tuning as a crucial factor also has implications for future LM development and training methodologies.

Clarity of Exposition: The paper is well-structured and generally clear in its presentation. The abstract effectively summarizes the key contributions, and the introduction sets the stage for the research questions. The use of benchmarks and quantitative metrics aids in understanding the results.

## Limitation or Weaknesses
Novelty: The concept of non-canonical tokenizations, while different from traditional adversarial examples (which often involve small, imperceptible perturbations), shares a thematic similarity in exploring how model inputs, even when semantically equivalent, can drastically alter performance due to underlying processing mechanisms. The paper could potentially build upon existing work more explicitly in certain areas. For example, some related work in adversarial examples or robustness to noise might be worth briefly connecting to the concept of non-canonical tokenizations, even if the focus is distinct. This could further contextualize the "surprise" factor of the findings.

Credibility with regard to Reproducibility: While the current information does not provide explicit details on open-sourcing code or datasets, the description of the methodology, including the use of 20 benchmarks and specific performance metrics, suggests a commitment to empirical rigor. The mention of "Our finetuning code was forked from allenai/open-instruct" indicates a reliance on publicly available resources, which could enhance reproducibility. However, without direct confirmation of shared code (I searched but couldn't find a fork from allenai/open-instruct for ) and data for the specific experiments conducted, full reproducibility remains to be explicitly verified.

## Overall Assessment
Good paper, accept

The paper presents a novel and significant discovery: instruction-tuned LLMs surprisingly handle non-canonical tokenizations well, even boosting performance on some tasks. Its methodology is sound, with evaluations across 20 benchmarks providing strong empirical evidence. The exposition is also generally clear.

However, the paper has critical weaknesses. It lacks deep contextualization with related research like adversarial examples, limiting its broader theoretical impact. The connection to model robustness to input noise, or even the theoretical underpinnings of tokenization as a form of data representation, remains superficial. This omission prevents the paper from fully articulating how its "broken tokens" represent a distinct, yet related, challenge or opportunity compared to other forms of input perturbations. Furthermore, it falls short in discussing the practical complexities, such as computational overhead, and potential negative impacts of implementing dynamic tokenization in real-world scenarios.

Finally, insufficient transparency regarding the open-sourcing of all specific code and datasets for its experiments raises reproducibility concerns. These shortcomings collectively reduce the paper's overall completeness and prevent a "Strong Accept" rating.

---

### Decision · Program_Chairs · 2025-06-10

Accept